# Fungal Resistance and Leaching Behavior of Wood Treated with Creosote Diluted with a Mixture of Biodiesel and Diesel

Kathleen Walker [1], Himadri Rajput [1], Alexander Murray [1], Glenn W. Stratton [1], Gordon Murray [2] and Quan (Sophia) He [1,*]



1   Faculty of Agriculture, Dalhousie University, Truro, NS B2N 5E3, Canada
2   Stella-Jones Inc., Truro, NS B2N 5C1, Canada
*   Correspondence: quan.he@dal.ca

**Abstract:** This study evaluated the effect of biodiesel as a co-solvent with the wood preservative creosote to reduce the amount of hydrocarbon-based carrier utilized. Small blocks of wood were treated at a pilot scale using three different creosote concentrations. The diluent used was a blend of 80% soybean biodiesel and 20% petroleum diesel. The efficacy of creosote was tested against brown rot and white rot fungi. The results of the wood-block test and agar test suggested that there was no significant effect of biodiesel on the efficacy of creosote as a wood preservative. As creosote-treated wood is commonly used for railway ties, its potential impact on the surrounding environment was also assessed by studying the leaching behavior of creosote–biodiesel–diesel blend treated railway ties. Rainfall simulators were used to imitate an exposure of treated wood to a significant amount of rainfall. Wood core drilled from the exposed railway ties and leaching water samples were analyzed for the levels of polycyclic aromatic hydrocarbons (PAHs) and total petroleum hydrocarbons (TPHs). Overall, this study demonstrated that the diluent containing biodiesel had no negative effect on the performance of creosote as a wood preservative and towards the natural environment.

**Keywords:** wood preservative; creosote; wood-decay fungi; biodiesel; leaching

## 1. Introduction

Wood has been used extensively due to its properties such as renewability, high impact strength, low thermal conductivity, low density, and high strength–weight ratio [1]. However, in view of its high biodegradability and susceptibility to attacks by insects or microorganisms (viruses, bacteria, and fungi), wood preservation is required to increase its durability and service life. Fungi can significantly degrade the wood, but viruses and bacteria have no critical effect on wood quality. Wood preservatives have been used since 1830, and they can extend the service life of wood up to 25–50 years. In the process of wood preservation, chemical or biochemical compounds are infused into the wood, which prevents its decay caused by fungi, insects, bacteria, algae, mildew, and other microorganisms [2]. Wood preservatives are broadly divided into two types, namely, organic and inorganic, and among them, creosote, chromium copper arsenate (CCA), and pentachlorophenol are the most common in Canada [3]. Creosote is an organic wood preservative isolated from coal-tar distillation processes and consists of a mixture of heterogeneous polycyclic aromatic hydrocarbons (PAHs). It is a widely used oil-borne wood preservative due to its excellent fungicide and insecticide behavior [4]. Creosote-treated wood has a lifetime of up to 30 years or even more and is mostly used for making utility poles, telephone lines, and railway ties [5,6]. For instance, untreated railway crossties have a service life of five years, whereas creosote-treated ties could last over 30 years [7]. However, when creosote-treated wood poles and railroad ties encounter rainwater, harmful compounds can leach into the surroundings [8]. Owing to indestructibility and non-degradability, the discharge of high-risk preservatives from treated wood gives rise to a serious threat to the environment.

Till now, the methods used for the detoxification of wood preservatives are mainly landfilling and combustion [9]. However, utilization of these methods offers a second pathway through which these contaminants can significantly enter the ecosystem. When landfilling is practiced, the wood preservatives can migrate to the soil and get adsorbed, ultimately making their way to the groundwater resources [10]. Alternatively, if combustion is applied to wood preservatives, it may lead to the release of toxic compounds into the atmosphere, ultimately contaminating the environment [11]. Hence, the development/modification of more efficient, low-cost, and eco-friendly methods for the utilization of wood preservatives, as well as the evaluation of their environmental impact, are highly desired.

Petroleum-derived hydrocarbons such as diesel are used as a carrier for wood preservatives and play a significant part in the performance of treated wood products. The efficiency of the carrier and its interaction with the wood preservative are believed to have impacts on the biocidal performance, depletion rate, and movement of the preservative across the wood structure [12–14]. However, these petroleum-derived hydrocarbons can further add to the issues of their non-renewable nature and treated-wood disposal. Hence, the utilization of biodiesel, a product from renewable sources as an alternative carrier or additive to petroleum diesel, is of current interest. Previous studies have shown that non-petroleum-based diesel exhibited lower toxicity and rapid biodegradability [15–17]. These properties are of interest for application in wood preservation due to biodiesel's high flashpoint and its relatively low environmental implications. However, there is very limited data available on the effect of biodiesel as a carrier on wood treatment efficiency and biodegradability of a wood preservative [18]. To date, no additional studies have been conducted involving co-solvents of biodiesel and diesel used as a diluent for creosote wood treatment. Similar research is also very limited, including a study on pentachlorophenol degradation in the presence of biodiesel conducted by Langroodi et al. (2012) [19] and an evaluation on the efficacy of copper naphthenate using biodiesel as a carrier performed by McKillop (2014) [20]. Hence, there is a need to explore the efficiency of biodiesel as a carrier for the preservative treatment of wood.

The idea behind this study is to reduce the amount of petroleum-based carriers used for creosote treatment without compromising the effectiveness of the preservative on wood protection. An additional benefit is the observed odor suppression effect of biodiesel, which helped provide an odorless, safer, and friendly environment for treaters and the surrounding neighborhood. Industrial interest is shifting towards the utilization of biodiesel-based carriers. The use of biodiesel at this level not only significantly reduces odor but also increases the flashpoint resulting in safer handling and storage of the preservative mixture [21,22]. Hence, this study is focused on the evaluation of the effect of a diluent containing biodiesel and diesel in the wood preservative creosote. The diluent was composed of 80% soybean biodiesel and 20% petroleum diesel. The wood samples were treated with creosote and diluent blend using a pilot-scale wood pressure treatment system. A fungal analysis was performed to investigate the effect of diluent on the efficacy of creosote against wood decay fungi. The leaching behavior of creosote was also studied to assess the potential environmental impact that the preservatives could have on surrounding ecosystems. Rainfall simulators were used to estimate leaching due to a significant amount of rainfall over a short period of time, and the environmental impact was studied. The outcomes of this study are expected to give an insight into the alterations that can be made in the wood preservation process and are particularly useful for developing greener and more environmentally friendly wood preservation processes.

## 2. Materials and Methods

### 2.1. Chemicals and Materials

The wood preservative used in this study was a P1/P3 Canadian-origin creosote provided by Ruetgers Canada Inc., Hamilton, ON, Canada. P1/P3 signifies that the creosote was a distillate derived from the carbonization of bituminous coal or a mixture of coal tar distillate and petroleum oil. Canada Clean Fuels, Brampton, Ontario, provided

a blend of 80% soybean biodiesel and 20% diesel (B80 ULSD # 2), which was used in this study as the diluent. All the American Type Culture Collection (ATCC) reference strains of fungi species used for the decay trials were purchased from Cedarlane, ON, Canada. The yeast and mold agar (YM agar) and Difco™ Potato Dextrose Agar used for fungal growth were obtained from Fisher Scientific. The wood samples were sent for sterilization to Nordion's Gamma Centre of Excellence, Laval, QC, Canada, and the irradiation lot number was 727602.

### 2.2. Pilot Plant Setup for Wood Preservative Treatment

The pilot plant used in this study (Supplementary Materials Figure S1) was designed as a miniature version of a full-scale wood pressure treatment system usually employed in wood treatment industries. It consisted of a 4 L holding tank where the preservative was stored, heated, and then transferred to the pressure vessel by a piston pump. The pump was also used to pressurize the pressure vessel. A pressure release valve and a pressure gauge were attached to a pipe from the pressure vessel, which dispensed the treating solution back into the storage tank, allowing the pressure to be reliably controlled. The main line had numerous drainage valves so that the system could easily be cleaned. The creosote and diluent were mixed in a graduated cylinder before being poured into the holding tank of the pilot plant, and an empty cell procedure without initial vacuum was applied. A cycle of 30 min pressure treatment to wood blocks at 150 psi was practiced, followed by a 30 min expansion bath. After loading wood samples as well as the creosote/diluent into the pilot plant, the temperature was increased to 93.3 °C (200 °F) to mimic full-scale operation procedures. The temperature of the expansion bath was maintained at 98.9 °C (210 °F). After the expansion bath, the treating cycle was complete, and the preservative solution was drained and stored in a cold, dark environment.

### 2.3. Fungal Decay Test

This study was conducted to compare the difference in the fungal decay among wood blocks treated with creosote with different concentrations, taking biodiesel as the co-carrier along with diesel. The agar block method was developed by referencing both Standard EN 113 (British Standards Institution, 1997) [23] and American Wood Protection Association (AWPA) Standard E10-16 (AWPA, 2016) [24].

2.3.1. Selection of Test Fungi and Its Inoculation

Five species of fungi (three brown-rot fungi and two white-rot fungi) were utilized for this study based on species listed in the AWPA and British Standard EN 113 responsible for wood decay. The brown rot fungi were *Rhodonia placenta (ATCC 11538-TT)*, *Neolentinus lepideus (ATCC 12653)*, and *Gloephyllum trabeum (ATCC 11539)*. The white rot fungi chosen for this study were *Stereum hirsutum (ATCC MYA-2819)* and *Trametes versicolor (ATCC 12679)*. *G. trabeum* (GT) and *R. placenta* (RP) were both chosen because of their common use in many experiments, *Trametes versicolor* (TV) for its rapid growth, *Neolentinus lepideus* (NL) for its tolerance to creosote, and *Stereum hirsutum* (SH) was chosen because it was one of the recommended white rot fungi in the AWPA and has been extensively included in previous research studies involving creosote [25]. The received frozen cultures of fungal species were revived and stored for experimentation. *R. placenta, N. lepideus, S. hirsutum*, and *T. versicolor* were cultured according to ATCC Medium 200 (YM agar/broth, 4.8%), while *G. trabeum* required ATCC Medium 337 (potato dextrose yeast agar).

2.3.2. Wooden Block Preparation and Decay Test Setup

A series of 50 × 25 × 7.5 mm blocks were milled from hardwood samples of rock maple (*Acer saccharum*) so that they would fit in an average-sized Petri dish. Many of the AWPA standards indicate the use of a non-durable coniferous wood such as southern yellow pine (SYP); however, rock maple was chosen for this study based on its common use in the rail tie industry. The treatment was applied to the wooden blocks as explained in

Section 2.2 with the following three preservative treatments: 100% creosote, 70% creosote + 30% diluent, and 50% creosote + 50% diluent. The term diluent refers to a solution of 80% soybean biodiesel and 20% diesel. When a percent creosote value is given, it refers to the amount of creosote present, and the balance would be the diluent. Ten replicates were used for each type of preservative treatment, and a group of control blocks (without preservative treatment) were set aside.

Before the fungal test, the treated wood blocks were first placed in an oven for one week at a temperature of 60 °C to stabilize the moisture content of the wood. Afterward, they were placed in a conditioning chamber for a total of 21 days with a temperature range of 20–30 °C ± 2 and relative humidity of 25%–50%. After conditioning, the blocks were weighed and then vacuum sealed and were sent for sterilization by gamma rays at 29.7 kGy and 32.2 kGy to inhibit any unwanted biological growth that could interfere with the decay experiments. Afterward, the blocks were aseptically weighed and placed on the agar plate (100 mm × 15 mm) with the fungi cultures (see Section 2.3.1 for the agar/fungi combination) and kept in an incubator at 25 °C ± 1 with a relative humidity of 50% for a 16-week period. To ensure that the culture transfer would be successful, they were given time to reach a size of 25 mm before the wood blocks were added to the dish. Once the wood blocks were placed in contact with the fungi, they were evaluated periodically, and observations were recorded. After the incubation period was completed, the wood was removed from the dish, and excess fungi were scraped using a scalpel. The mass of the wood block was recorded, and it was then placed in a conditioning room for a total of 21 days with a temperature range of 20–30 °C ± 2 and relative humidity of 75%. Here, the least-squares mean (LS) was used to determine which treatments were significantly different from the control.

### 2.4. Agar Study for Testing the Efficacy of Creosote

An additional agar study was carried out to test the efficacy of creosote with diluent by following the procedure reported by Guillén et al. (2009) [26]. This experiment had the same preparation requirements described in Section 2.3.2, except that wood was not used, and 5 replicate experiments were performed for each fungal strain. This serves as the second method to evaluate the efficacy of creosote with a diluent containing biodiesel. It has been reported that *N. lepideus* (NL), *G. trabeum* (GT), and *R. placenta* (RP) are in the top 10 most prevalent fungi found on softwoods [27] and *T. versicolor* (TV) along with *G. trabeum* (GT) are very common on hardwoods. In this study, blank paper disks (Oxoid Antimicrobial Susceptibility Test Discs, CT0998 B) were dipped in three different blends of creosote and placed on the agar plates equidistant from the fungi after they had grown to the size of 25 mm in diameter. The plates underwent a 16-week incubation period at a temperature of 25 °C, and later observations were recorded.

### 2.5. Leaching Test

Rainfall simulation was used in this study to determine the amount of leaching associated with a given amount of rainfall over time in a controlled environment. The AWPA Standard E11-16 was adapted and modified to plan the leaching experiment [28]. The results are expected to indicate the long-term performance of wood preservatives. The full-size rail ties (Supplementary Materials Figure S2) were treated with a 70% creosote + 30% diluent blend in an industrial-scale plant like the pilot-scale unit described in Section 2.2. After being treated, the ties were left in a dry room until they were cut to the dimensions 100.3 cm × 15.24 cm × 20.32 cm, so they would properly fit in the rainfall simulators used in this study. The cut ends were sealed with Paraffin wax to prevent leaching from the cross-section that was cut. The total area exposed to rainfall was calculated to be 5094.5 cm$^2$ and explained in detail in Supplementary Figure S3 and Supplementary Materials text S1.

The simulators were constructed using plastic containers, a pump, and nozzles for dispersing water over the wood sample, as shown in Supplementary Figure S4. This allows for several years of environmental exposure to be re-created in a short period of time.

In addition, the use of a closed-loop system prevented the need for a large volume of water. There were eight rainfall simulations completed using six rainfall simulators. Before reuse, the simulators were cleaned between different experimental runs. The total volume available in each simulator was 378 L. To ensure that the wood was above the water level and was not submerged during the experiment, 55 L of water was added at the beginning. The water was recirculated through the rainfall simulators at a rate of 4.13 L/min. During the experiment, wood was exposed for 9 h a day for 5 days duration (totaling 45 h), and the total rainfall that fell on the wood was 11,151 L. The experiment simulated 14.2 years of total rainfall (details are provided in Supplementary Materials text S2 and Equations (S1)–(S4)) (https://www.currentresults.com/Weather-Extremes/Canada/wettest-cities.php [Last Accessed 9 September 2022]) [29]. After completion of the experimental duration, both drilled wood cores from the railway ties and leachate water samples were analyzed by GC-MS for semi-volatile organics and hydrocarbons.

### 2.6. Analytical Methods

#### 2.6.1. Measurement of Wooden Block Decay over Time

At the end of the 16-week incubation period, as discussed in Section 2.3.2, the blocks were removed from the incubator, and all visible mycelia were carefully removed by scraping the fungal mass off the wood block with a razor blade and then brushing it with a small brush. The blocks were then weighed and placed in the incubator, which was set at 60 °C and ~15% humidity to standardize the moisture content. After 7 days, they were re-weighed and placed in a conditioning chamber for 21 days and then re-weighed for the final time. The weight loss percentage due to fungal activity was calculated using an equation provided by the AWPA (2016) [24] as follows:

$$\text{Mass loss (\%)} = 100 \times (T3 - T4)/T3$$

where T3 is the mass of the wood before it was placed on the agar with the fungi, and T4 is the final mass of the wood block after conditioning or oven.

#### 2.6.2. Statistical Analysis

The statistical software used in this study was SAS 9.4 and Minitab 2017. Fungal decay and leaching experiments used different statistical analyses; however, a 95% confidence interval ($\alpha = 0.05$) was set for both. The fungal decay experiment, as explained in Section 2.3, was analyzed as a two-factor factorial. The factors were the preservative concentration at four levels (0%, 50%, 70%, and 100%) and the species of fungus at five levels. In addition, the concentration of creosote was also compared through multiple means comparison using Tukey's test. Statistical analyses were completed for each percent weight loss calculated in the decay studies. The leaching study explained in Section 2.5 was analyzed using a two-sample *t*-test in Minitab. Since wood cores and water samples were taken before and after the experiment, a two-sample *t*-test was completed to determine if the data were significantly different in terms of PAH and total petroleum hydrocarbon (TPH) leaching.

### 3. Results and Discussion

#### 3.1. Efficacy of Creosote Evaluated by Wood Blocks

All wood blocks were treated with solutions with creosote concentrations of 50%, 70%, and 100%, as described in Section 2.2. The treatability is depicted in Figure 1. The range of retention was between 8 and 25 pcf, where retention of the untreated control group remained zero. It was observed that the net retention of creosote increased as the percentage of creosote in the solution increased. This is understandable as the wood block up-took more creosote ion treating solutions with higher creosote concentrations.

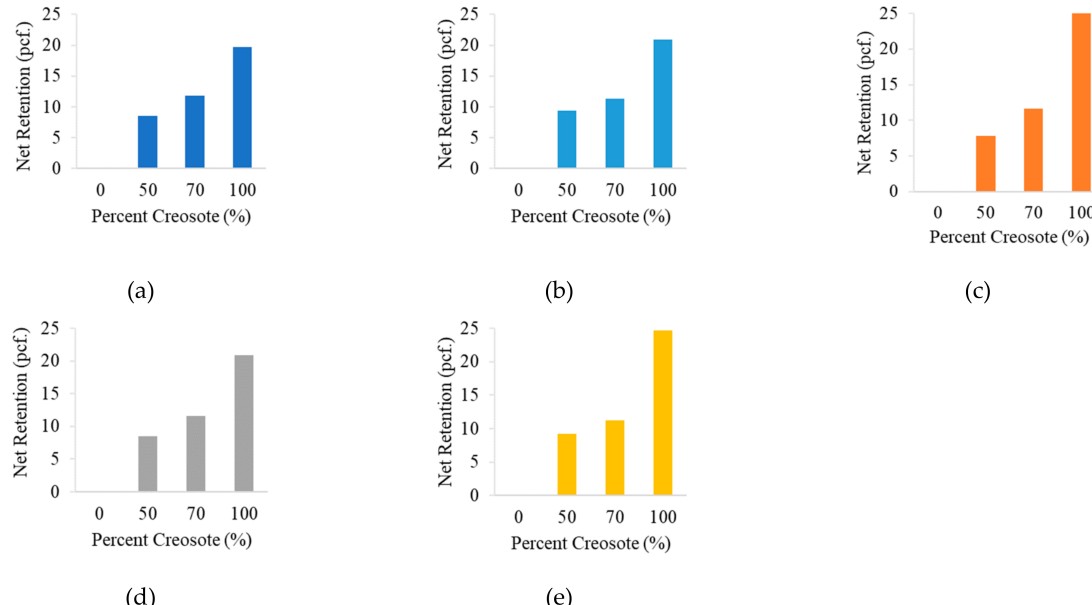

**Figure 1.** The net retention (pcf) of creosote in wood blocks treated at different solutions for testing the efficacy of five fungal species: (**a**) GT, (**b**) NL, (**c**) RP, (**d**) SH, and (**e**) TV.

The efficacy of wood blocks treated with different solutions was evaluated by following AWPA Standard E10-16. The decay resistance of test blocks was determined by the percent loss of wood mass after they had been exposed to the decaying fungi for a 16-week incubation period, as explained in Section 2.3. A two-factor factorial was employed to determine whether the diluent in the creosote or the fungal species had a significant effect on the mass loss. Here, one factor was the creosote concentration of treating solution at four levels, 0% creosote +100% diluent, 50% creosote +50% diluent, 70% creosote +30% diluent and 100% creosote +0% diluent. The other factor was the fungi species, including three brown rot fungi *Rhodonia placenta* (RP), *Neolentinus lepideus* (NL), and *Gloephyllum trabeum* (GT), and two white rot fungi, *Stereum hirsutum* (SH) and *Trametes versicolor* (TV). The results are shown in Table 1.

**Table 1.** Mass loss of wood blocks after exposure to decaying fungi for 16 weeks and two-factor factorial analysis using LS means.

| Fungi Species | Creosote Concentration (%) | Mass Loss (%) | Standard Deviation | Mass Loss (%) LS Means * | LS Means Multiple Means Comparison ** |
|---|---|---|---|---|---|
| GT | 0 | 12.13 | 4.67 | 12.12 | B |
| GT | 50 | 5.86 | 0.73 | 5.86 | D |
| GT | 70 | 5.99 | 0.21 | 5.99 | D |
| GT | 100 | 5.96 | 0.41 | 5.97 | D |
| NL | 0 | 9.71 | 1.79 | 9.71 | BC |
| NL | 50 | 6.42 | 0.47 | 6.47 | D |
| NL | 70 | 5.95 | 0.88 | 5.95 | D |
| NL | 100 | 6.05 | 0.305 | 6.05 | D |
| RP | 0 | 7.33 | 3.57 | 7.11 | CD |
| RP | 50 | 5.94 | 0.27 | 6.005 | D |
| RP | 70 | 6.28 | 2.99 | 6.36 | D |
| RP | 100 | 6.27 | 0.19 | 6.27 | D |
| SH | 0 | 4.35 | 1.01 | 4.23 | D |
| SH | 50 | 6.03 | 0.19 | 6.06 | D |
| SH | 70 | 6.15 | 0.34 | 6.12 | D |
| SH | 100 | 6.07 | 1.04 | 6.07 | D |
| TV | 0 | 17.27 | 4.15 | 17.26 | A |
| TV | 50 | 6.90 | 1.45 | 5.93 | D |
| TV | 70 | 6.81 | 0.61 | 6.81 | D |
| TV | 100 | 6.75 | 0.43 | 6.74 | D |

* LS mean is not the same as the true mean; for the true mean, refer to the descriptive statistics in Table 1. ** Data with the same letter are not significantly different at $\alpha$ = 0.05.

As seen in Table 1, the wood blocks treated by only diluent without any creosote had a significant mass loss due to fungal activity; for example, 12.13% for GT, 9.71% for NL, 7.33% RP, and 17.27 for TV. This is understandable, as diesel and biodiesel just acted as a carrier and had no pesticide effect. Surprisingly, for the fungal species SH, the test block treated without any creosote showed a mass loss of 4.35% and similar degrees of decay of 6.03%, 6.15%, and 6.07% for the test blocks treated with creosote. This might be because the fungus SH has grown the slowest out of all the species examined. It has been observed that the test species SH generally attacks spruce trees [30], while here, the wood block samples were made from rock maple. Examining the mass loss with respect to the retention, the mass losses are comparable for all five fungal species tested in this study. For example, in fungal species GT, the mass loss was 5.86%, 5.99%, and 5.96%, corresponding to retentions of 8.54 pcf, 11.8 pcf, and 19.68 pcf, respectively. Other species showed the same trend. This implies that the threshold of protection has been achieved, and higher retentions are not economical and not necessary either since the protection performances are similar at the retention levels tested in this study.

Further statistical analysis was conducted to estimate the means for a balanced population [31], and the results are presented in Table 1. Tukey's test was also completed within each fungal species to determine if there was a significant difference based on the mean percent weight loss of the wood, as shown in Table 2. All statistics were completed where $\alpha = 0.05$, with a confidence interval of 95%. As observed, there was a significant difference in mass loss between control blocks and the creosote-treated blocks in the two-factor factorial analysis where $\alpha = 0.05$ (Table 1). However, there was no significant difference in mass loss among blocks treated with 100% creosote, 70% creosote, or 50% creosote. This indicated that neither the species of fungi nor the concentration of creosote showed a significant difference in the efficacy of wood samples against fungi decay. This was also confirmed through Tukey's test (Table 2).

**Table 2.** Multiple means comparison with Tukey's test of wood block weights *.

| Fungi Species | Creosote Concentration (%) | Tukey's Test ** |
| --- | --- | --- |
| GT | 0 | A |
| GT | 50 | B |
| GT | 70 | B |
| GT | 100 | B |
| NL | 0 | A |
| NL | 50 | B |
| NL | 70 | B |
| NL | 100 | B |
| RP | 0 | A |
| RP | 50 | A |
| RP | 70 | A |
| RP | 100 | A |
| SH | 0 | A |
| SH | 50 | A |
| SH | 70 | A |
| SH | 100 | A |
| TV | 0 | A |
| TV | 50 | B |
| TV | 70 | B |
| TV | 100 | B |

* Each fungus was tested separately. ** Data with the same letter are not significantly different at $\alpha = 0.05$.

The results of the fungal decay analysis found that the diluent did not have a significant effect on the efficacy of creosote in preventing the decay of the wood blocks. Thus, it can be concluded that adding biodiesel as a co-carrier in wood-preserving solutions had no negative effect on the efficiency of wood preservation.

### 3.2. Efficacy of Creosote Evaluated by Agar Test

The test fungi were also exposed to creosote treatments using agar plates without wood blocks. The plates were incubated for a 16-week period as that in the wood blocks test (Section 3.1) and then removed to observe the interactions between the fungi and the preservative blends, such as the zone of inhibition and fungal growth.

Figure 2 is a close-up image of fungus NL in an agar test and serves as an example showing how to evaluate the zone of inhibition as well as what is meant when the fungi are said to have grown equally across the plate. As seen, the zone of inhibition was visible for the 70% creosote disc, which refers to the small circle around the creosote-dipped disc. There was also equal fungi growth throughout the plate, evidenced by some concentrated growth around the 50% creosote and 50% diluent. The observed result is not surprising because NL is known as a creosote-resistant species.

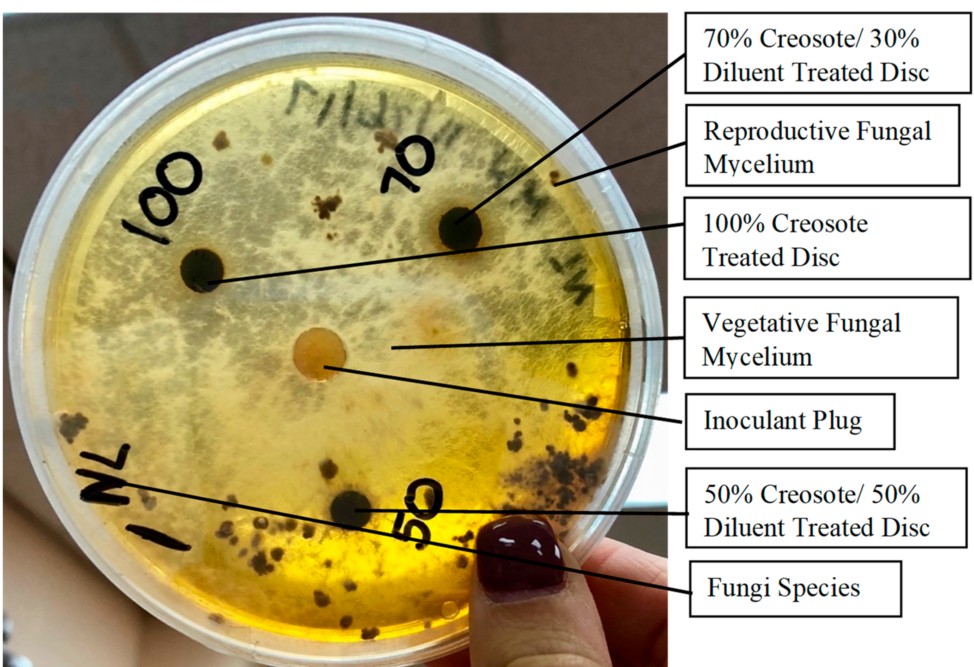

**Figure 2.** Close-up image of the fungus *N. lepideus* (NL) to fully explain the terms used in the description of the plates.

Figure 3a–e show the agar test results of five fungi, GT, NL, RP, SH, and TV, respectively. For fungus GT, as shown in Figure 3a, there were similarly small zones of inhibition around each preservative-dipped disc, indicating that the diluent did not affect the interaction between the fungi and the preservative. The fungus NL interacted with the 50% creosote disc; however, it did not react with the 70% creosote disc or the 100% creosote disc, as shown in Figure 3b. The response of RP can be observed in Figure 3c, showing no observational differences, and it is acceptable to state that either 50%, 70%, or 100% creosote was effectively resistant to the decay fungi RP. The agar plate test results from fungus SH show that it did not interact with any of the discs (Figure 3d). Furthermore, the fungus grew equally away from all the discs indicating that it did not favor any preservative treatment. This is understandable since SH is a slow-growing fungus. The fungus TV is known for its rapid growth and can be observed in Figure 3e. It is evident that there was no significant growth in any direction on the fungi plates. There were zones of inhibition around each of the creosote treatments, which indicated that regardless of the concentration of diluent added, the creosote was able to prevent fungal growth.

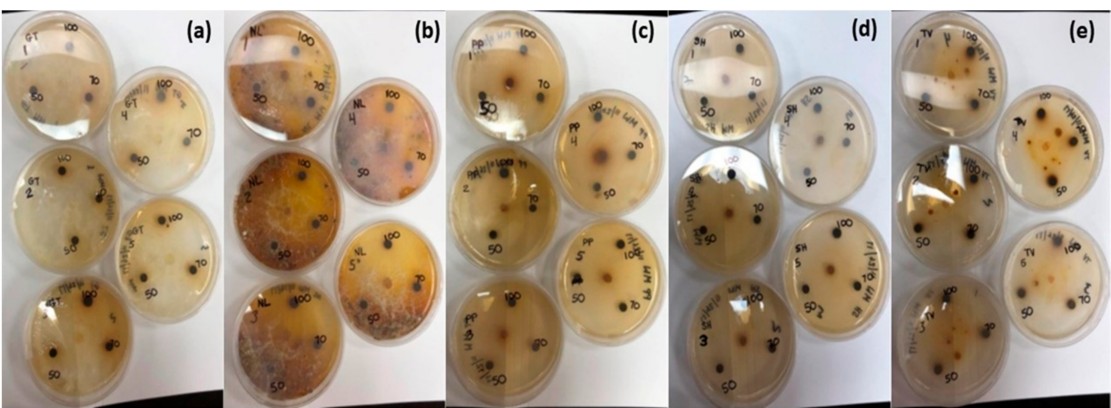

**Figure 3.** Blank discs with 100%, 70%, and 50% creosote discs on agar with (**a**) *G. trabeum* (GT), (**b**) *N. lepideus* (NL), (**c**) *R. placenta* (RP), (**d**) *S. hirsutum* (SH), and (**e**) *T. versicolor* (TV).

Overall, based on the results using two evaluation methods (wood block and agar disc), the addition of biodiesel did not impact the efficacy of creosote against all the five fungi species tested in this study. The results are in accordance with the observations reported previously, where no loss in the efficiency of wood preservatives (pentachlorophenol or copper naphthenate) was observed when biodiesel was used as a co-solvent [19,20].

### 3.3. Leaching Study of Creosote-Treated Wood

The rainfall simulators were used to expose the rail ties, and both wood core and water samples were collected before and after rainfall simulation experiments and analyzed for the presence of TPHs and total PAHs originating from creosote. The concentrations of PAHs and TPHs in wood core and water before and after leaching experiments are summarized in Tables 3 and 4, respectively.

**Table 3.** Concentrations of PAHs and TPHs in wood cores before and after leaching.

| Sample | Concentration (mg/kg) | Standard Deviation | Sample Size |
|---|---|---|---|
| PAHs Before | 39,088 | 10,171 | 8 |
| PAHs After | 34,583 | 8311 | 8 |
| TPHs Before | 8396 | 1902 | 8 |
| TPHs After | 7699 | 2006 | 8 |

**Table 4.** Concentrations of PAHs and TPHs in water samples collected in duplicate after leaching.

| Sample | Average Concentration (mg/kg) | Standard Deviation | Sample Size |
|---|---|---|---|
| PAH Sample 1 | 8.206 | 2.720 | 8 |
| PAH Sample 2 | 8.756 | 2.674 | 8 |
| TPH Sample 1 | 57.28 | 16.94 | 8 |
| TPH Sample 2 | 58.09 | 16.39 | 8 |

A two-sample *t*-test was applied to examine the significance of the statistical analysis of the obtained wood core samples. For PAHs, the *t*-value and *p*-value were −0.97 and 0.350, respectively, and for TPHs, the *t*-value and *p*-value were 0.71 and 0.488, respectively. Based on the two-sample *t*-test, there was no significant difference between the concentration of PAHs or TPHs before and after the leaching experiment at $\alpha = 0.05$. For the water samples, they were collected in duplicate, and the concentrations of PAHs and TPHs were determined (Table 4). Two two-sample *t*-tests were performed for samples before and at the completion of the leaching experiment. For PAHs, the *t*-value and *p*-value were −0.41 and 0.690, respectively, and for TPHs, the *t*-value and *p*-value were −0.10 and 0.942, respectively. The statistical analysis indicates that there is no significant difference between Sample 1 and Sample 2 for either the PAH or TPH concentrations at $\alpha = 0.05$.

The standard set by the Canadian Council of Ministers of the Environment (CCME) Environmental quality guidelines, the permissible exposure limit for PAHs in freshwater

sediment is 7.38 mg/kg of dry weight [32]. As observed from Table 4, the average PAH concentration in the water was 8.206 mg/kg, which is higher than the standard stated above. However, since the experiment was simulated for a period of approximately 14.2 years, the overall leaching would theoretically be lower than 7.38 mg/kg. Furthermore, the RBCA (2016) [33] states that the acceptable limit of TPHs in sediment ecological screening levels for the protection of freshwater and marine aquaculture life should be a maximum of 500 mg/kg. Based on the results from Table 4, it is evident that the values are much lower than this, which indicates that the standard was met. It is important to acknowledge that in this study, the leaching experiment was simulating a long period of time in which the TPH would not leach into the environment at once but over a period of 14.2 years. Hence, it can be concluded that the rail ties treated with 70% creosote + 30% diluent blend leach at rates that meet environmental standards in Canada.

## 4. Conclusions

This study has evaluated the effect of the addition of a diluent composed of 80% soybean biodiesel and 20% petroleum diesel on the effectiveness of creosote as a wood preservative. The efficacy of creosote against five specific wood-decaying fungi was estimated. It was concluded that the addition of biodiesel did not have a negative effect on the protective ability of creosote against wood rot fungal species. The recommended blend of creosote and diluent for wood treatment is 70% creosote and 30% diluent. As observed from the fungi analysis, the blend of 70% creosote and 30% diluent was most beneficial against fungi because it acted like the 100% creosote, and it did not have additional fungi growth like the 50% creosote/50% diluent. Leaching tests also showed that the environmental regulations for PAH and TPH were met when the preservative solution was made up of 70% creosote and 30% diluent. Further research work can be explored to test the efficacy of creosote/biodiesel/diesel-based treating solution on other wood-decaying fungal species.

**Supplementary Materials:** The following supporting information can be downloaded at: https://www.mdpi.com/article/10.3390/f14030625/s1, Figure S1: Pilot plant used to treat wood blocks; Figure S2: Rail ties treated with 70% creosote + 30% diluent blend used for rainfall simulation before they were cut to smaller pieces; Figure S3: Box design of rail tie with dimensions 100.3 cm × 20.32 cm × 15.24 cm after being cut for the leaching experiments; Figure S4: Rainfall simulator where the water is being pumped through the PVC pipes and raining onto the wood sample sitting above the pool of water.

**Author Contributions:** K.W.: conceptualization, methodology, investigation. H.R.: writing—original draft preparation. A.M.: data curation, investigation. G.W.S.: validation, writing—review and editing. G.M.: funding acquisition, validation. Q.H.: supervision, writing—review and editing. All authors have read and agreed to the published version of the manuscript.

**Funding:** Authors are thankful to Stella-Jones Inc. for funding this project and providing the facilities required for the successful completion of this project. NSERC funding (CRDPJ 4921179-15) is also acknowledged.

**Data Availability Statement:** The datasets used and/or analyzed during the current study are available from the corresponding author upon reasonable request.

**Conflicts of Interest:** The authors declare no conflict of interest.

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
