# Peer review of "Fungal Resistance and Leaching Behavior of Wood Treated with Creosote Diluted with a Mixture of Biodiesel and Diesel"

_forests, doi:10.3390/f14030625_

Round 1

Reviewer 1 Report

Dear authors 

Your fungal decay test results for controls are unfortunately invalid. As you know and as indicated in EN 113 standard test method, If mass loss is under 20% the decay test is invalid. Also you did not include control sample not  treated with any solution. Since decay tests are the main skeleton of the article and the decay tests are invalid in this study, I cannot recommend the article to be published.

Line 33: Microbes include virus, bacteria and fungi. Fungi can degrade wood but virüs and bacteria have no critical effect on wood. That is why you can use insects and fungi here.

Line 37. Sorry I could not  find the reference you cited related with bacteria can decay wood.

Line 135: indicate at least one citation for the previous research studies you said.

Line 138: Mention about concentration/percent of agar media. It should be 4.8% in EN 113.

Line 154. How it is possible that you keep them in relative humidity  ranged 25-75%. It seems is not a stable.

Line 155. You kept the treated woods in an incubator at 25 °C ± 1 for a 16-week period. What was the relative humudity ? again 25-75%?

Line 166. Is there a special reason for keeping the humidity range high? (from 25 to 75).

Line 176. Which standart did you used for the leaching procedure? You can add to the text.

Line 204. “The blocks were then weighed and placed in an oven for 7 days at a temperature 207

of 60 °C to stabilize the moisture content” (What did moisture indicate ?  high or low?, because if you recorded low moisture content, fungi probably did not Show efficiency. Your decay results (controls) also showed that fungi species degraded limited mass

Line 244. Table 1. What do you mean with sample size 10,10,10, 9, 10, 8 !!? Replica?

Line 256-261: These sentences are not finding or results, they are material and methods sentences.

Line 258. You should have carried out decay control samples from the woods that were not treated with any preservative or solution. Here is the major deficience in your study. 9.71% mass loss was occurred in NL control and 6.47% in 50% creosote treated. There is only about 3% difference between them.  In this case, how can you understand that biodiesel or diesel solution protected wood or not against fungi? because there is no untreated samples in the study. Even if you used biodiesel treated one as control, mass losses were found lower than 20% that is why decay test is invalid according to EN 113 standart test method.!!!. The reason of the occurening low mass loss under 20% was probably related with applying low relative humudity 25! -75%

Line 281-290. Unnecessary information because not results maybe move to MM section

Line 326-335 is not result sentences they were already explained in MM section.

Table 1 and Table 2 can be combined.

Author Response

Reviewer 1:

Your fungal decay test results for controls are unfortunately invalid. As you know and as indicated in EN 113 standard test method, If mass loss is under 20% the decay test is invalid. Also, you did not include control sample not treated with any solution. Since decay tests are the main skeleton of the article and the decay tests are invalid in this study, I cannot recommend the article to be published.

Response: see #12 below. The authors fully agree with the comment of the reviewer. The EN113 standard requires mass loss greater than 20%. However, we just adapted the method from EN113 as a guideline to design our experiments to evaluate the effect of biodiesel as a carrier on the efficacy of the new wood preservatives formulations as well as on leaching (concerns over enhanced leaching by biodiesel).  EN113 is used to determine the toxic threshold value, which is not our research objectives. Hence, mass loss lower than 20% is not very relevant in this case as our purpose is different. We apologize that we have not made ourselves clear. We have changed accordingly in introduction and methodology sections.

The authors are thankful to the reviewer for his considerable efforts towards improving the quality of this manuscript and highlighting the issues which needed our attention. Following is the specific response towards the queries raised by the reviewer.

  1. Line 33: Microbes include virus, bacteria, and fungi. Fungi can degrade wood but virüs and bacteria have no critical effect on wood. That is why you can use insects and fungi here.

Response: Thanks for bringing this up. We have revised accordingly as added in line 33 and 34.

  1. Line 37. Sorry I could not find the reference you cited related with bacteria can decay wood.

Response: The authors reflect on the error made and more appropriate reference (Line 39) is now added which explains the bacterial wood decay (Chapter 10, 10.2.5).

  1. Line 135: indicate at least one citation for the previous research studies you said.

Response: The citation is now added as reference 23 and the reference numbering is updated.

  1. Line 138: Mention about concentration/percent of agar media. It should be 4.8% in EN 113.

Response: The percentage is now added in Line 142.

  1. Line 154. How it is possible that you keep them in relative humidity ranged 25-75%. It seems is not a stable.

Response: we apologies that we have not made it clear.  We have clarified in section 2.3.2. Here in conditioning room the relative humidity is 25-50%. The relative humidity was not well controlled in our previous condition room at that time. Now we have a new conditioning room.

  1. Line 155. You kept the treated woods in an incubator at 25 °C ± 1 for a 16-week period. What was the relative humidity? again 25-75%?

Response: Here the relative humidity in incubator is 50%. It was well controlled by humidifiers and dehumidifiers.

  1. Line 166. Is there a special reason for keeping the humidity range high? (from 25 to 75).

Response: Here the relative humidity was ab 75%. The reason is to mimic full-scale parameters. Especially rail ties, a significant number of them are in humid areas.

  1. Line 176. Which standard did you used for the leaching procedure? You can add to the text.

Response: The standard used to design the experiment is now added in the text and reference list (reference no. 28).

  1. Line 204. “The blocks were then weighed and placed in an oven for 7 days at a temperature of 60 °C to stabilize the moisture content” (What did moisture indicate ? high or low?, because if you recorded low moisture content, fungi probably did not Show efficiency. Your decay results (controls) also showed that fungi species degraded limited mass.

Response: The incubator was set at 60C and ~15% humidity to standardize the moisture content.

  1. Line 244. Table 1. What do you mean with sample size 10,10,10, 9, 10, 8 !!? Replica?

Response: Yes, the numerical digit signifies the number of replicates for each condition. We set up 10 replications; but there are a few outliers, giving replication only 8 or 9. We understand this is bit confusing. We have removed these numbers. Thanks 

  1. Line 256-261: These sentences are not finding or results, they are material and methods sentences.

Response: Thanks; They are now moved to materials and methods section (Line 168-169).

  1. Line 258. You should have carried out decay control samples from the woods that were not treated with any preservative or solution. Here is the major deficience in your study. 9.71% mass loss was occurred in NL control and 6.47% in 50% creosote treated. There is only about 3% difference between them. In this case, how can you understand that biodiesel or diesel solution protected wood or not against fungi? because there is no untreated samples in the study. Even if you used biodiesel treated one as control, mass losses were found lower than 20% that is why decay test is invalid according to EN 113 standard test method.!!!. The reason of the occurring low mass loss under 20% was probably related with applying low relative humidity 25! -75%.

Response: The authors fully agree with the comment of the reviewer. The EN113 standard requires mass loss greater than 20%. However, we just adapted the method from EN113 as a guideline to design our experiments to evaluate the effect of biodiesel as a carrier on the efficacy of the new wood preservatives formulations as well as on leaching (concerns over enhanced leaching by biodiesel).  EN113 is used to determine the toxic threshold value, which is not our research objectives. Hence, mass loss lower than 20% is not very relevant in this case as our purpose is different. We apologize that we have not made ourself clear. We have changed accordingly in introduction and methodology sections.

  1. Line 281-290. Unnecessary information because not results maybe move to MM section.

Response: Thanks, these are low level mistakes. We apologized that PIs did not spot these. The sentences are moved to materials and method section (Line 174-178).

  1. Line 326-335 is not result sentences they were already explained in MM section.

Response: The irrelevant sentences are removed as per the suggestion (Line 327).

  1. Table 1 and Table 2 can be combined.

Response: Table 1 and 2 are combined as suggested.

Reviewer 2 Report

The paper is based on evalution of using biodisel as carrier for adding to creosote and for evaluation the influence on efficacy.

The scientific novelty is not so huge.

I put comments and correction directly in pdf file

Author Response

Reviewer 2:

The authors are thankful to the reviewer for the raised queries and given suggestion for the improvement of this manuscript. All the suggestions are incorporated in revised file and the response is provided below.

Response:

Thank the reviewer for his or her efforts in improving the quality of this manuscript . The authors highly appreciated;  revisions are done as follows:

  1. Line 33, the term “microbes: is changed to “microorganisms”.
  2. Line 39, “in Canada” is added now as per the reviewer’s suggestion.
  3. Line 45, “a serviceable life” is now replaced with “service”.
  4. Line 128, “Postia placenta” is replace with “Rhodonia placenta” and corrected throughout the manuscript as highlighted in red.
  5. The updates standard is now cited (reference 23).
  6. Line 249, representation of names of fungi is corrected throughout the manuscript.
  7. Table 1, “Weight (%)” now replaced with “Mass loss (%)”.

Reviewer 3 Report

Although the paper is well-written and the experiment is quite interesting, it is however, not acceptable at the present form.   

Introduction should be improved by providing a clearer problem statement, justification and objectives.

Materials and methods section is clearly written.

Line 240-241 – please italicize scientific name of fungi

The experiments in the materials section are lacking. In my opinion, to warrant publication in a Q1 journal, the author should add:

1.      Treatability of the wood (weight percent gain and polymer retention etc after treatment). It is very important as the chemicals reside in the wood is an important factor for its biological resistance.

2.      FTIR – what is the reaction between the wood and the impregnant

3.      Fungi resistance of the wood after leaching treatment. It is insufficient for the authors to test only the leaching rate of the treated wood. Its resistance after leaching is a more important criteria to observe.

Author Response

Reviewer 3:

Although the paper is well-written and the experiment is quite interesting, it is however, not acceptable at the present form. Materials and methods section is clearly written. The experiments in the materials section are lacking. In my opinion, to warrant publication in a Q1 journal, the author should add:

The authors are thankful to the reviewer for his time and efforts towards our manuscript. The raised concerns are addressed below:

  1. Introduction should be improved by providing a clearer problem statement, justification, and objectives.

Response: Thanks for helping improve the quality of the manuscript. Clarifications are added on the raised concerns in Line 80-85 and supporting references are also added.

  1. Line 240-241 – please italicize scientific name of fungi.

Response: Done.

  1. Treatability of the wood (weight percent gain and polymer retention etc after treatment). It is very important as the chemicals reside in the wood is an important factor for its biological resistance.

Response: Thanks a lot for the valuable comments. Though our main objective in this study is to evaluate the difference in the wood protection between biodiesel/diesel and pure diesel as carriers, we agree it is better to include information of treatability.  The retention data are added in the Table 1.  

  1. FTIR – what is the reaction between the wood and the impregnant.

Response: The authors agree that FTIR study will give us scientific insight of interaction of wood and the impregnant, however, our current study is more practically oriented and revolved around the recommendation for industrial use of biodiesel-based carrier as an alternative. Our approach for this study is different but surely FTIR analysis and other characterization technologies can be explored in the future to understand the interactions.

  1. Fungi resistance of the wood after leaching treatment. It is insufficient for the authors to test only the leaching rate of the treated wood. Its resistance after leaching is a more important criteria to observe.

Response: These are great points from the reviewer, The authors cannot agree more; however, our main aim in this study is to find out whether addition of biodiesel affects/increases the leachability of treated wood, and it came out that it did not. This is a positive sign as wood treatment industries are moving towards the utilization of biodiesel as carrier dur to odor and safety issues with petroleum-based carriers. Although, we really liked the idea of testing resistance of wood towards decay after the leaching experiments and this will be included in our next study.

Round 2

Reviewer 1 Report

Dear authors 

After a deep review, I have changed the decision to publishing  since the primary aim of your work is to develop biodiesel as a carrier. However, your fungal resistance test results are still controversial. In your future work, please apply a constant relative humidity of min 65% in the incubator as specified in the standard test method (EN 113). 

Good luck.

Author Response

Thanks for the reviewer's encouraging words. Sure in our future study, we will follow strictly EN113 and set a constant relative humidity of min 65%  which will allow us not only to observe the difference between diluents applied , also determine toxic threshold. 

Reviewer 3 Report

I understand the constraints raised by the authors where some experiments can't be done. However, I do hope that the authors can at least plot the graph between the relationship of retention and weight loss, and discuss the trend.

Author Response

Thank the reviewer for his or her efforts to improve the quality of the manuscript.

Treatability has been added as Figure 1 , and relevant discussion is added and highlighted in red.